# Safety Survey on Lone Working Magnetic Resonance Imaging Technologists in Saudi Arabia

**DOI:** 10.3390/healthcare11050721

**Published:** 2023-03-01

**Authors:** Sami A. Alghamdi, Saad A. Alshamrani, Othman I. Alomair, Yazeed I. Alashban, Abdullah H. Abujamea, Essam H. Mattar, Mohammed Almalki, Mohammed Alkhorayef

**Affiliations:** 1Radiological Sciences Department, College of Applied Medical Sciences, King Saud University, Riyadh 4545, Saudi Arabia; 2Department of Radiology and Medical Imaging, College of Medicine, King Saud University, Riyadh 4545, Saudi Arabia

**Keywords:** MRI safety regulations, lone MRI workers, American College of Radiology (ACR), Royal Australian and New Zealand College of Radiologists (RANZCR), magnetic resonance safety officers (MRSO)

## Abstract

Purpose: The American College of Radiology (ACR) requires MR personnel not to work alone due to the increased risk of safety issues such as projectiles, aggressive patients, and technologist fatigue. As a result, we intend to assess the current safety of lone-working MRI technologists in MRI departments in Saudi Arabia. Materials and Methods: A cross-sectional study using a self-report questionnaire was conducted in 88 Saudi hospitals. Results: A response rate of 64% (174/270) was obtained among the 270 MRI technologists which were identified. The study discovered that 86% of MRI technologists had prior experience working alone. In terms of MRI safety training, 63% of MRI technologists received such training. A question about lone MRI workers’ awareness of the ACR’s recommendations revealed that 38% were unaware of such recommendations. Furthermore, 22% were misinformed, believing that working alone in an MRI unit is optional or depends on the individual’s desire to work alone. Working alone has the primary consequence of being statistically significantly associated with projectile/object-related accidents/mistakes (*p* = 0.03). Conclusion: Saudi Arabian MRI technologists have extensive experience working alone without supervision. Most MRI technologists are unaware of lone working regulations, which has raised concerns about accidents/mistakes. There is a need for MRI safety training and adequate practical experience to raise awareness of MRI safety regulations and policies related to lone working among departments and MRI workers.

## 1. Introduction

Magnetic resonance imaging (MRI) is a non-invasive imaging technique which is used to map the interior structure and function of different body parts [1]. A strong magnetic field and non-ionizing electromagnetic radiation are used in MRI [2,3]. Unfortunately, MRI is not completely safe, as its properties raise safety issues and enhance the risk of harm to unskilled staff and patients in the MRI setting [1,4]. The Emergency Care Research Institute (ECRI) released its annual report in 2020, and MRI was ranked 8th on its list of the top 10 health device hazards due to the risks associated with missing implant data, which can put the patient in danger and create MRI scan delays [5].

A strong magnetic field is associated with risks such as temporary biological effects, torque, and projectiles, whereas radiofrequency (RF) pulses may cause tissue heating and induce a current in implanted devices, thus necessitating the calculation of the specific absorption rate (SAR) [6,7]. A rapid switching gradient, on the other hand, may cause peripheral nerve stimulation (PNS), and a high level of acoustic noise may cause hearing loss [1,8,9] In response to these safety concerns, the American College of Radiology (ACR) has provided updated recommendations and guidelines regarding different aspects of MR safety (such as magnetic resonance safety officers (MRSO) who have the proper training to make sure safety practices are followed in MRI units) to keep everyone in the MRI area safe [10].

Two MRI staff members should be present in zones II (the patient screening zone) through IV (the MRI scanner room) and in the scanning room at all times, as recommended by the American College of Radiology (ACR) and the Royal Australian and New Zealand College of Radiologists (RANZCR). Concerns regarding worker and patient safety arise when employees are required to work alone in an MRI environment without nearby support, making it more difficult to obtain assistance in the event of an emergency involving the patient or the equipment. As an example, when MRI technologists work alone, they face issues such as inaccurately filled-out questionnaire forms, trouble dealing with anxious and hostile patients, a missing object that increases the projectile risk, quenching, electrical hazards, and crises [11,12,13].

Lone workers are classified in a number of ways. The Canadian Centre for Occupational Health and Safety (CCOHS) says “a person is alone at work when they are on their own: when they cannot be seen or heard by another person and when they cannot expect a visit from another worker”. According to the Alberta Occupational Health and Safety Code, “a worker is recognized as working alone if the worker works alone at a worksite in circumstances where assistance is not readily available when needed in the case of an injury, illness, or emergency” [9]. Furthermore, “a worker may be considered to be working alone if there is a reasonable assumption that a call for help will not or cannot be answered, and that the worker’s absence will not be discovered for some time” [14,15,16].

To the best of our knowledge, no official Saudi government-funded studies have been conducted on this aspect. To evaluate lone work and its potential dangers, this study examines the existing state of affairs for MRI technologists in Saudi Arabia.

## 2. Materials and Methods

### 2.1. Ethical Approval

In accordance with the recommendations of the Ministry of Health’s Institutional Review Board (IRB) at King Fahad Hospital in Al-Baha (Education, Training Center and Academic Affairs Scientific and Research Committee), approval to conduct this study was granted on 7 November 2021 (IRB No. 03112021/2). Each participant was given an explanation of their rights as a free person taking part in this experiment. Each participant was also given access to the study’s rationale and the researcher’s contact details.

### 2.2. Online Study

An online questionnaire was distributed to 270 MRI technologists between November and December 2021 for this cross-sectional study. This study included MRI technologists of both genders with varying levels of qualification and at least six months of experience performing MRIs, and their consent to participate was obtained. The study included 88 hospitals in Saudi Arabia, both private and public. This work builds on our previously published method. Because it is an extended research study, it is a national Saudi study rather than a regional one, as in our previous paper [17].

### 2.3. Questionnaire Survey

The questionnaire was primarily based on a survey adopted from previously published research [13,16,17,18]. There was a total of 20 questions across 6 different categories. There were some yes/no questions, some multiple-choice questions, and some questions based on a five-point Likert scale. The participants were asked about their demographic information, their prior lone-worker experience, the department’s resources, their familiarity with the ACR’s lone-worker regulations, their MRI safety training, the MRI unit’s efficiency for lone MRI technologists, the impact of this work on their confidence, their preference for working with other MRI technologists, and their exposure to safety-related accidents and mistakes. The following conditions warranted omission: problems including MRI technologists with less than six months’ experience, partially completed surveys, and reluctant participants.

The General Department of Radiology of the Ministry of Health distributed the electronic questionnaire link to all radiology officials in all of the regions and then to the supervisors of MRI units in government hospitals in the Kingdom of Saudi Arabia. The questionnaire was distributed by the supervisors to the technicians that were targeted in this study. In the case of private hospitals, we called the MRI unit supervisors and asked them to take part in the survey.

### 2.4. Statistical Analysis

SPSS software (version 26) was used for data coding, tabulation, and analysis (Armonk, NY, USA: IBM Corp.). The chi-squared (χ^2^) and Fisher’s exact tests were used to analyze the quantitative data (numbers and percentages) to determine the associations between the variables. Means and standard deviations (means ± SD) were used to summarize quantitative data, and the Mann–Whitney and Kruskal–Wallis tests were applied to examine differences between the groups if the independent variables did not follow a normal distribution. For statistical significance, a *p* value of less than 0.05 was used (*p* < 0.05).

## 3. Results

Of the 270 MRI technologists identified in Saudi Arabia, 47 did not respond to the survey questionnaire, and 49 were excluded (experience of less than six months), granting a final response rate of 64% (174/270; 62.6% male and 37.4% female).

The data from the descriptive analysis of the participants’ demographic information are displayed in Table 1. Most of the participants had between 4 and 10 years’ worth of MRI experience (25.1%), and the vast majority (83%) of the MRI technologists were employed by public hospitals. A typical day for an MRI technologist involves treating 10–16 patients. The participants’ work information is shown in Table 2. Although the majority of the respondents (68.4%) reported that their departments lacked a dedicated magnetic resonance safety officer (MRSO), nearly 80 percent (79.9%) said that protocols for reporting safety issues in the MRI unit were made available to them. Eighty-three percent of the surveyed MRI technologists had first-aid certification, and 63 percent had completed MRI safety training. The majority of the MRI technologists (39%) reported high levels of confidence in their ability to perform their duties in the presence of other medical staff members (e.g., nurses), whereas only 4% reported poor levels of confidence. Table 2 also displays the percentage of MRI technologists that indicated that they would want to collaborate with other MRI technologists.

In regard to the statement “According to the ACR, working alone in the MRI unit is optional, depending on the person’s desire and ability to work alone,” the results showed that 38% did not know about this regulation, and 22% were wrongly informed, whereby they agreed with the idea that working alone in the MRI unit is optional. In contrast, 40% were aware of the regulations prohibiting MRI technologists from working alone in MRI units (Table 2).

Concerning the relationship between awareness of the ACR’s recommendations, gender, qualifications, and experience of the lone MRI technologist, the results revealed that the level of awareness was statistically significantly correlated with qualifications and years of experience in the MRI department (*p* = 0.00 and *p* = 0.015, respectively; Table 3). According to the Mann–Whitney test results, there is a significant difference in qualifications. The awareness differences between diploma and master’s degree holders (*p* = 0.255 and *p* = 0.003, respectively) were more significant in comparison to the bachelor’s and master’s degree holders. Furthermore, there was a significant difference in awareness between those with 1–3 years of experience and those with more than 10 years of experience (*p* = 0.050). The level of MRI safety awareness did not differ significantly between genders. To summarize, MRI technologists with bachelor’s and master’s degrees and more than 10 years of experience were more aware of the regulations governing lone work (Table 3).

The findings revealed that lone MRI technologists (*n* = 149) and MRI technologists working with another medical staff member (*n* = 155) work daily “occasionally” (45% and 46%, respectively) (Figure 1). Table 4 compares the workplaces and the rates of MRI technologists working alone or with other staff members.

Over 86% (149 out of 174) of MRI technologists in Saudi Arabia have substantial expertise working independently. Yet another prevalent scenario for MRI techs is to work alongside another member of the medical team (such as a nurse) (89%, 155/174). There is a statistically significant difference (*p* = 0.002) in the rate of lone working between businesses (private and public). The daily rate of lone working among MRI techs in private hospitals is higher than among those in public hospitals (mean rank = 112.38 vs. 82.52, respectively). In Table 5, we see a statistically significant correlation between the percentage of MRI technologists who work alone and the frequency with which accidents involving projectile items (e.g., magnets) occur (*p* = 0.03).

Concerns (the fear of being subjected to aggressive behavior by the patient and their relatives) and accidents/mistakes (feelings of tiredness and exhaustion, which may affect safety) are significantly lower when MRI technologists work with another medical staff member present (*p* = 0.027 and *p* = 0.00, respectively).

The presence of an MRSO is correlated with the existence of safety accident reporting policies, as shown in Table 6.

## 4. Discussion

Previous published work has been restricted to the southern region of Saudi Arabia [17]. Our most recent study is a comprehensive study that includes the entire country (13 regions). It is a cross-sectional study that collected data from 270 MRI technologists in 88 Saudi hospitals using a self-report questionnaire. As a result, this work is a continuation of our previously published method, which was carried out in 23 hospitals and collected data from 79 technologists. Different MRI machines use different designs and offer a wide variety of magnetic field strengths to achieve better performance with a greater signal-to-noise ratio (SNR) compared to other methods [18]. However, MRI is not entirely risk-free, and complications might arise if an inexperienced or lone MRI technologist is allowed to work in an MRI environment without the constant supervision of medical professionals [1,13,19].

When it comes to MRI, having a dedicated MRSO on site and MRI safety training programs are crucial [11,20]. The majority of the respondents (68.4%) reported that their departments lacked a dedicated magnetic resonance safety officer (MRSO) and that they had received less instruction in MRI safety than they had in first aid. Previous research shows this to be true; the vast majority of the southern Saudi Arabian participants said that their MRI centers lacked an MRSO. Only around half of the MRI technologists had participated in formal MRI safety training programs. According to the report, this is because the Ministry of Health in Saudi Arabia requires neither an MRSO to be present nor MRI safety training. Having mechanisms in place to monitor MRI safety, report and assess unit hazards, and avoid further events is crucial. We found that most institutions have procedures in place for reporting MRI safety issues. This finding is in line with what we found in the southern region in our earlier study [17,21].

The findings presented in this study show that lone working is common among Saudi MR technologists. Eighty six percent (149 out of 174) of the respondents indicated that they have had this experience. In addition, the participants lacked enough familiarity with the ACR’s rules prohibiting lone work in an MRI unit. Most of the respondents believed that working alone in an MRI unit was permissible, indicating that they were either unaware of the rules or were given false information. Table 3 demonstrates that awareness was negatively correlated with a lack of education and experience working in an MRI unit. Since radiographers who only have a diploma have very limited theoretical training and no clinical experience, it stands to reason that they would be woefully unprepared to deal with the hazards associated with using an MRI machine. These results go against the grain of a different study [17,21]. Accidents and blunders are more likely to happen when MRI technologists work alone. As our research shows, most MRI technologists would rather work in a team setting. This research confirms what Dewland et al. [13] found to be true in the region of Western Australia, where there is a marked preference for solitary labor. For reasons related to patient well-being, the MRI technologists in the study preferred to collaborate with other trained MR professionals. This makes sense considering that some MRI techs have poor perceptions of lone working and are seeking laws to ban this practice [15,22,23].

Working alone presents a number of safety risks that are not present while working alongside other medical staff members. As the findings of this study indicate, concerns (such as a fear of being subjected to aggressive behavior by the patient and their relatives) are significantly lower when MRI technologists work with another medical staff member present (*p* = 0.027). This finding is consistent with our previous study, which found that MR technologists feel such concerns when working alone [24]. In addition, our findings show that working alone is generally associated with accidents/mistakes, such as forgetting some questionnaire questions or forgetting to complete the patient’s weight registration. Forgetting to record a patient’s weight may affect patients’ safety because of the specific absorption rate (SAR) that must stay in the normal range to avoid overheating and burning and is required to calculate how much contrast media to use [1,15,23,24].

To the best of our knowledge, there are no local safety instructions or guidelines related to working alone in MRI units. Therefore, it is strongly recommended for rules and regulations to be established in order to eliminate lone working and thus improve MRI safety for both patients and healthcare workers. 

The primary shortcoming of this study is that it depended on a self-administered questionnaire rather than on the collection of actual incidents. This is because certain arrangements or agreements are needed to collect these forms of data, which is difficult due to certain constraints in hospitals’ or institutions’ regulations and procedures [11,12]. In conclusion, MR technologists are discouraged from working independently by both the RANZCR and the ACR. Our research shows that the majority of MRI technologists in Saudi Arabia (86%) have substantial experience working alone. As a result, it is hoped that this research will serve to educate the Saudi clinical community and, more specifically, the business sector.

## 5. Conclusions

The conclusion of this study confirms that MRI technologists in Saudi Arabia have a large amount of experience working alone without other medical staff. Most MRI technologists are unaware of international regulations on lone working, which has resulted in an increase in concerns regarding accidents and mistakes. There is a demand for training in MRI safety and adequate practical experience to increase the awareness of departments and MRI workers regarding MRI safety regulations and policies related to lone working.

## Figures and Tables

**Figure 1 healthcare-11-00721-f001:**
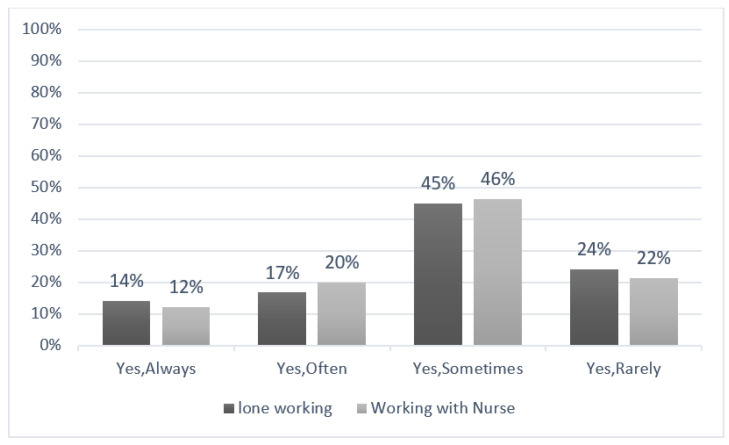
Relationship between days spent working alone and the frequency of working in the presence of another medical staff member.

**Table 1 healthcare-11-00721-t001:** Participants’ demographic data.

Items	Criteria	Number (%)
Experience	Less than 6 months	49 (22)
	From 6 months to less than a year	31(13.9)
	1–3 years	43 (19.3)
	4–10 years	56 (25.1)
	More than 10 years	44 (19.7)
Participants from each region	Albaha	8 (5)
	Asir	13 (7)
	Riyadh	42 (24)
	Makkah	31 (18)
	Eastern Province	22 (13)
	Northern Borders	7 (4)
	Jezan	13 (7)
	Tabuk	6 (3)
	Qasim	14 (9)
	Madinah	11 (6)
	Jawf	2 (1)
	Najran	2 (1)
	Hail	3 (2)
Nationality	Saudi	157 (90.23)
	Non-Saudi	17 (9.77)
Gender	Male	109 (62.6)
	Female	65 (37.4)
Age	Less than 30 years.	53 (30.5)
	Between 30–40 years.	91 (52.3)
	More than 40 years.	30 (17.2)
Average number of patients/machine/days	Less than 10 (<10) patients	31 (17.8)
	10–16 patients	74 (42.5)
	More than 16 (>16) patients	69 (39.7)
Type of health care setting	Public hospital	145 (83)
	Private center	29 (17)
Job type	Full-time	165 (95)
	Part-time	9 (5)
Qualifications	Diploma	29 (13)
	Bachelor’s degree	104 (46.6)
	Master’s degree	35 (15.7)
	PhD	5 (2.2)
	Saudi fellowship for radiology technologists	1 (0.4)

**Table 2 healthcare-11-00721-t002:** Participants’ work information.

Items	Criteria	Number (%)
The presence of an MRSO in the department	Available	55 (31.6)
	Not available	119 (68.4)
The availability of policies for the reporting of safety accidents that occur in the MRI department	Available	139 (79.9)
	Not available	35 (20.1)
Have you received training in the following areas: [MRI safety]?	Yes	109 (63)
	No	65 (37)
Have you received training in the following areas: [first aid]?	Yes	145 (83)
	No	29 (17)
The extent of self-confidence and preferences when you are working as the only MRI technologist in the presence of another medical staff member (such as nurses)	Not confident at all	5 (3)
	Not very confident	7 (4)
	Moderately confident	33 (19)
	Highly confident	61 (35)
	Completely confident	68 (39)
According to the American College of Radiology (ACR), working alone in the MRI unit is optional, depending on the person’s desire and ability to work alone	True	38 (22)
	False	69 (40)
	I don’t know	67 (38)
Preference to work with another qualified MRI technologist	Very undesirable	1 (1)
	Undesirable	10 (6)
	Neutral	24 (14)
	Desirable	58 (33)
	Very desirable	81 (47)

**Table 3 healthcare-11-00721-t003:** Participants’ awareness in comparison to gender, qualifications, and experience.

Q. According to the American College of Radiology (ACR), Working Alone in an MRI Unit is Optional, Depending on the Person’s Desire and Ability to Work Alone				
Variable	Criteria	Number of Incorrect Answers (%)	Number of Correct Answers (%)	*p*-Value
Gender	Male	24 (63.2)	85 (62.5)	0.979 ^a^
	Female	14 (36.8)	51 (37.5)	
	Diploma	18 (13)	11 (29)	
Qualification	Bachelor	80 (59)	24 (63)	^b^
	Master	32 (24)	3 (8)	0.255 ^c^
				0.003 ^d^
	Ph.D.	5 (3)	0 (0)	
	Fellowship	1 (1)	0 (0)	
MRI experience	6 months to <1 year	23 (17)	8 (21)	
	1–3 years	38 (28)	5 (13)	0.015 ^e^
	4–10 years	40 (29)	16 (42)	0.050 ^f^
	>10 years	35 (26)	9 (24)	0.527 ^g^

^a^ Fisher’s exact test; ^b^ difference between total qualifications and awareness according to the Kruskal–Wallis test; ^c^ awareness between diploma and master’s according to the Mann–Whitney U test; ^d^ awareness between bachelor’s and master’s according to the Mann–Whitney U test; ^e^ difference between total experience and awareness according to the Kruskal–Wallis test; ^f^ awareness between 1–3 years and more than 10 years according to the Mann–Whitney U test; ^g^ awareness between 4–10 years and >10 according to the Mann–Whitney U test.

**Table 4 healthcare-11-00721-t004:** Comparison between the workplaces and the rates at which MRI technologists work alone or in the presence of other staff members.

Based on Four-Point Likert Scale Questions ^a^	Public (Mean Rank)	Private (Mean Rank)	H ^b^	*p*-Value
Rate of lone working (*n* = 149)	82.52	112.38	9.163	0.002
Rate of working as the only MR technologist with the presence of another medical staff member (such as nurses) (*n* = 155)	87.06	89.72	0.074	0.785

^a^ The four-point Likert scale question ranges from (rarely to always). ^b^ Test statistics of the Kruskal–Wallis test.

**Table 5 healthcare-11-00721-t005:** Concerns and safety accident/mistake events experienced by MRI technologists.

Comparison between Different Concerns and Accidents/Mistakes experienced by MRI Technologists while They Are Working as: (A) A Lone MRI Technologist (*n* = 149) (B) An MRI Technologist in the Presence of Another Medical Staff Member (such as Nurses) (*n* = 155)			H ^a^	*p*-Value
Concerns ^b^	i. Regarding patient safety	A	2.387	0.496
		B	4.51	0.211
	ii. Regarding the safety of the persons present in the MRI unit	A	3.363	0.339
		B	1.606	0.658
	iii. Any accidents that may affect safety	A	4.737	0.192
		B	6.651	0.084
	iv. Fear of being subjected to aggressive behavior by the patient and their relatives	A	1.711	0.635
		B	9.165	0.027
	v. How to deal with the patient’s anxiety and fears	A	5.63	0.131
		B	0.908	0.823
	vi. Fear of feeling isolated	A	3.018	0.389
		B	3.553	0.314
Accidents/Mistakes ^c^	i. Forgetting the patient’s weight registration	A	2.40	0.49
		B	3.53	0.32
	ii. Forgetting some questions in the patient safety questionnaire	A	2.98	0.39
		B	1.51	0.68
	iii. Projectile objects (e.g., keys, pens, scissors, and hairpins)	A	9.15	0.03
		B	2.98	0.39
	iv. Feelings of tiredness and exhaustion that may affect safety	A	6.17	0.10
		B	18.70	0.00

The concerns of MRI technologists when they are working alone (A) or with another medical staff member such as a nurse (B) regarding different safety issues. ^a^ Kruskal–Wallis test (H). ^b^ Each item is based on a five-point Likert scale from 1 = not at all concerned to 5 = extremely concerned. ^c^ Each item is based on a five-point Likert scale from 1 = never to 5 = always.

**Table 6 healthcare-11-00721-t006:** The relationship between the presence of an MRSO in the department and the availability of policies for safety accident reporting.

Descriptive Statistics				
	Mean	Std. Deviation	*n*	Correlation
The presence of an MRI safety officer (MRSO) in the department	1.68	0.466	174	0.310
The availability of policies for reporting safety accidents that occur in the MRI unit	1.2	0.402	174	

**Notes: Abbreviations:** MRI, magnetic resonance imaging; MRSO, magnetic resonance safety officer.

## Data Availability

Data is unavailable due to privacy or ethical restrictions.

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
