# Peer review of "Safety Survey on Lone Working Magnetic Resonance Imaging Technologists in Saudi Arabia"

_healthcare, 2023, doi:10.3390/healthcare11050721_

Round 1
Reviewer 1 Report
The manuscript entitled “Safety Survey on Lone Working Magnetic Resonance Imaging Technologists in Saudi Arabia” by Alghamd et al. reports on a Safety Survey in Saudi Arabia about MRI technologists that work alone.
It is worth remarking that this Reviewer does not have specific experience or data about MRI in Saudi Arabia, so please take the review within this framework.
The manuscript includes valuable information for healthcare.
Some issues are listed below:
3. Results:
1. Figure 1 quality must be improved.
2. Although the texts can be well followed and understood, a whole grammar spelling check might benefit.
Author Response
We would like to thank Reviewer 1 for taking the effort to review the manuscript.
We sincerely appreciate all your valuable comments and suggestions, which helped us in improving the quality of the manuscript.
"Please see the attachment"

Reviewer 2 Report
The authors have investigated several safety aspects in relation to MRI. The manuscript is well-written, with clear aims and an excellent introductory section. The manuscript covers an essential subject for the medical imaging community. I have a minor comment in relation to the discussion that the authors should address prior to the publication.
The discussion currently reads as a review of what other studies have reported, which is interesting and important, but they are not clearly linked back to the results.
Author Response
We would like to thank Reviewer 2 for taking the effort to review the manuscript.
We sincerely appreciate all your valuable comments and suggestions, which helped us in improving the quality of the manuscript.
"Please see the attachment"

Reviewer 3 Report
Good paper!
The technical details are OK. The statistics are well performed. I would like to see more details on the clinical significance of your results. Do they have any consequences on the patients' wellness or safety? What suggestions do you have for the medical authorities in your country in order to improve this situation? Otherwise, the paper looks fine.
Author Response
We would like to thank Reviewer 3 for taking the effort to review the manuscript.
We sincerely appreciate all your valuable comments and suggestions, which helped us in improving the quality of the manuscript.
"Please see the attachment"

Reviewer 4 Report
My comments ar attached as word format.

Author Response
We would like to thank Reviewer 4 for taking the effort to review the manuscript.
We sincerely appreciate all your valuable comments and suggestions, which helped us in improving the quality of the manuscript.
"Please see the attachment"
